# Exploring the immune responses triggered by vaccine formulations containing the recombinant *Schistosoma mansoni* 14kDa fatty acid-binding protein

**Poliane Silva Maciel**[iD]**, Gregório Guilherme Almeida,
Gardênia Braz Figueiredo de Carvalho, Rosiane A. da Silva-Pereira,
Lis Ribeiro do Valle Antonelli, Cristina Toscano Fonseca**[iD]*

Laboratório de Biologia e Imunologia de Doenças Infecciosas e Parasitárias, Instituto René Rachou, Fundação Oswaldo Cruz, Belo Horizonte, Minas Gerais, Brazil

* cristina.toscano@fiocruz.br

## Abstract

Many different *Schistosoma* antigens have been evaluated as vaccine candidates, including the recombinant form of the *Schistosoma mansoni* 14-kDa fatty acid-binding protein (rSm14). However, recombinant proteins often lack intrinsic immuno-stimulatory activity, a limitation that can be addressed by using vaccine formulations that contain adjuvants. In this work, we describe the immune response triggered by rSm14, a vaccine candidate against schistosomiasis currently under clinical trial, formulated with either (i) Monophosphoryl Lipid A (MPLA), (ii) MPLA/Alum, or (iii) Freund's adjuvant. rSm14/MPLA and rSm14/MPLA/Alum formulations induced increased frequency of effector and memory CD4$^+$ T and central memory CD8$^+$ T cells, respectively. Both formulations induced significant production of rSm14-specific IgG and IgG1 antibodies, which could recognize the protein's native form. The rSm14/Freund's formulation elicited a robust immune response characterized by increased levels of IFN-γ, TNF, IgG, IgG1, and IgG2c antibodies, and expansion of memory B cell. These soluble factors have been implicated in the efficacy of Sm14-based vaccines. Despite inducing both humoral and cellular immune responses, the different formulations did not impact worm burden and the number of eggs trapped in the liver and intestine. Altogether, our findings indicate a limitation in the use of the molecules assessed in this study, such as IFN-γ, TNF, and specific antibodies, as correlates of protection and vaccine efficacy.

## Introduction

Schistosomiasis is one of the most important neglected tropical diseases, having a significant impact on public health [1]. Over the last few decades, various *Schistosoma* antigens (Ag) have been evaluated as vaccine candidates, among them, the

---

**Data availability statement:** All relevant data are within the manuscript and its Supporting Information files. Raw flow cytometry data and Elisa plate readers are available at the ArcaDados Fiocruz repository under the following URL: https://doi.org/10.35078/BADAEY.

**Funding:** This study was financed in part by Coordenação de Aperfeiçoamento de Pessoal de Nível Superior-CAPES – Finance Code 001, Conselho Nacional de Desenvolvimento Científico e Tecnológico–Brasil (Grant Nos. 306188/2022-8 and 315540/2023-0), Fundação de Amparo a Pesquisa de Minas Gerais - Rede De Pesquisa em Doenças Infecciosas Humanas e Animais do Estado de Minas (RED00313-16), Fundação de Amparo a Pesquisa de Minas Gerais - Rede Mineira de  Imunobiológicos (RED00067-23), Fundação de Amparo a Pesquisa de Minas Gerais - Rede Mineira de Investigação em Mucosas e Pele (RED00096-22) and Fundação de Amparo a Pesquisa de Minas Gerais (APQ-03113-24, APQ-02481-23). The funders had no role in study design, data collection and analysis, decision to publish, or preparation of the manuscript.

**Competing interests:** The authors have declared that no competing interests exist.

*Schistosoma mansoni* 14-kDa fatty acid-binding protein (Sm14) [2]. Sm14 has a predicted physiological role in lipid uptake during schistosome development, since the parasite cannot synthesize fatty acids *de novo* and is thus dependent on the host as a source of lipids [2,3].

Recombinant proteins often lack intrinsic immunostimulatory activity, a limitation that can be addressed with vaccine formulations that contain adjuvants and with different antigen delivery systems [4,5]. Adjuvants (from the Latin "adjuvare" meaning "to aid" or "to help") were initially described as "substances used in combination with a specific antigen that produced a more robust immune response than the antigen alone" [6]. The effector mechanisms of adjuvants include: (1) sustained release of antigen at the site of injection, known as the "depot" effect; (2) improvement of antigen delivery, and enhancement of antigen processing and presentation by antigen-presenting cells (APC); and (3) up-regulation of cytokine and chemokine production [7–9]. Adjuvants, by triggering components of the innate immune response, alter the quality of the adaptive immune response [10,11].

The ability of the Sm14 to protect against schistosomiasis was initially demonstrated by Tendler et al (1996) [12] who showed that vaccination with a recombinant form of this protein alone or formulated with Freund's adjuvant (FA) induced 50–68% protection in Swiss mice and 89% in white rabbits. Fonseca et al. (2004) [13] observed that co-administration of exogenous IL-12 enhanced protective immunity engendered by rSm14 from 25 to 42.2% in C57BL-6 strain, which was dependent on IFN-γ and TNF production. In a DNA vaccine strategy, the *sm14* gene induced 40% of protection against *S. mansoni* infection and the production of IFN-γ and specific anti-rSm14 IgG antibodies in C57BL-6 mice [14]. Swiss mice immunized with rSm14 antigens in association with MPL-TDM Ribi adjuvant, showed 52% to 55% of protection [15]. In human phase I clinical trials, rSm14 was formulated with the adjuvant glycopyranosyl lipid A (GLA-SE) [16,17], which was subsequently used in Phase IIa (NCT03041766) and IIb (NCT03799510) clinical trials conducted in adults and children, respectively, in an endemic region of Senegal. However, according to the reports from clinical trials (n° NCT05658614), the immune response induced by the Sm14 vaccine was transient [16]. Furthermore, the association between the immune response profile and protection is not easily observed in humans, due to the sensitivity limitations of direct diagnostic techniques, associated with the absence of a controlled infection model that mimics natural infection, with the presence of oviposition. Additionally, the only study that evaluated the Sm14 response in preclinical tests, associated with an adjuvant similar in composition to that used in human, did not evaluate the immune response triggered by the vaccine formulation [15,18]. The Bilharvax® (recombinant Sh28GST/Alum) vaccine trial demonstrates the importance of establishing a good correlate of protection for schistosomiasis [19,20]. This vaccine advanced to Phase 3 efficacy studies without having an accepted clinical endpoint or a surrogate for protection, making the vaccine efficacy evaluation a difficult task [21]. Therefore, experimental approaches that explore the induction of the immune response triggered by different adjuvants, evaluating this response in the context of a challenge infection, are essential to assess the immunogenicity profile associated with protection.

In this study, we evaluated the immunogenicity of a recombinant Sm14 formulated with different adjuvants in C57BL-6 mice. The rSm14/Freund's formulation induced the production of IFN-γ and TNF, while the rSm14/MPLA/Alum formulation led only to an increase in IL-6 production. All formulations induced significant levels of specific IgG, with differences observed in antibody titers and profiles. Although the tested formulations induced distinct cellular and humoral responses, they failed to induce protection against challenge infection.

## Methods

### Mice and parasites

C57BL/6 female mice (6–8 weeks old) were obtained from the animal facility of the Instituto René Rachou (IRR)/FIOCRUZ-MG. Mice were maintained in a specific pathogen-free condition. The Ethics Committee approved all procedures performed on the animals on the Use of Laboratory Animals of Fiocruz (license numbers LW2/18 and LW13/20). The cercariae (LE strain) of *Schistosoma mansoni* were obtained from infected snails by exposing them to light for 1–2 hours to induce parasite shedding at the Lobato Paraense mollusk facility at the IRR. Before mice infection, cercarial numbers and viability were determined by light microscopy. Adult worms of the same *S. mansoni* strain were also obtained from infected mice and used to prepare the soluble *S. mansoni* worm antigen preparation (SWAP) [22] which was used in the enzyme-linked immunosorbent assays (ELISA).

### Expression and purification of the recombinant protein Sm14

The DNA sequence corresponding to the *S. mansoni* rSm14 coding region (Accession ID P29498) was used to construct a synthetic gene for expression in *Escherichia coli*. As previously described [18], to improve the structural stability of the Sm14 protein over time, the cysteine residue present in Sm14 at position 62 was replaced by valine. A codon optimization for the expression in bacteria was performed. The synthetic *sm14* gene, inserted between the restriction sites for the enzyme EcoRI and XhoI in the pET21b (+) multiple cloning site, was commercially acquired (GenScript, USA). rSm14 was expressed and purified using a C-terminal histidine tag, as previously described [23]. Briefly, the protein was expressed in *Escherichia coli* BL21 (DE3) carrying the plasmid construct pET21b/Sm14. Gene expression was induced with 1mM IPTG for 4 hours. After bacterial lysis under denaturing conditions, rSm14 was purified by affinity chromatography using the QIAexpress Ni-NTA Fast Start Kit (Qiagen, Germany). The purified protein was then dialyzed against phosphate-buffered saline (PBS) at pH 7.2, and its concentration was determined using the BCA Protein Assay Kit (Thermo Fisher Scientific, USA).

### Recombinant protein evaluation

A single rSm14 batch was used in all immunization trials. Its quality and purity were assessed by SDS-PAGE. Western blotting was performed to confirm his-tag recognition, as previously described [23] and the identity of the purified protein was confirmed by mass spectrometry. For this purpose, 15 µg of protein was separated by 15% SDS-PAGE and stained with Coomassie Brilliant Blue G-250 (Bio-Rad). All bands observed in the purified protein sample were separately excised from the gel, reduced using 10mM dithiothreitol (DTT), alkylated using 55mM iodoacetamide, and submitted to gel digestion using 12.5 ng/µL of trypsin (Trypsin Gold, Promega, USA) diluted in 50mM ammonium bicarbonate at 37°C for 16 hours. The peptide samples obtained were then extracted using acetonitrile (ACN) and trifluoroacetic acid (TFA) solution: 30% ACN and 3% TFA in the first two elution cycles, and 100% ACN and 3% TFA in the next two cycles. The peptides were concentrated using a SpeedVac (Thermo Scientific,USA) and desalted in reverse phase micro-columns ZipTip C18 (Millipore, Germany), according to the manufacturer's instructions. The peptides were then separated by capillary liquid chromatography, ionized by a nanoelectrospray source, and analyzed by tandem mass spectrometry (LC-MS/MS) in a nLC Ultra 1D Plus chromatograph (Eksigent) coupled to a LTQ Orbitrap XL ETD mass spectrometer (Thermo Scientific,

USA). Mass spectra were analyzed using the MaxQuant program (version 1.6.3.4). A search for the peptide fragments identified by mass spectrometry was performed against the WormBase ParaSite (version WBPS14) database, with trypsin defined as the enzyme used for protein digestion. Methionine oxidation and *N*-terminal protein acetylation were specified as variable modifications, and cysteine carbamidomethylation as a fixed modification. For identification of the peptides, at least seven amino acids were required. To evaluate conformational stability of the recombinant protein, 50 µg of rSm14, diluted in Tris-HCl 50mM buffer pH7,5, containing 2mM $CaCl_2$, were subjected to trypsin digestion (0.5 µg). Aliquots of 15 µL were collected at 1, 2, 5, 10, 20, 40, and 60 minutes. Digestion was stopped by the addition of protein sample buffer containing 10% β-mercaptoethanol. Aliquots of rSm14 without the addition of trypsin were used to normalize the data (T0), while heat-denatured protein (100°C) subjected to trypsin digestion was used as a positive control for proteolysis. Results were analyzed on a 15% SDS-PAGE gel stained with Coomassie Brilliant Blue G-250 (Bio-Rad). Densitometry analyses were performed at ImageJ, and results are reported as the percentage of intact protein over time. This assay was performed twice with the same protein batch at a digestion temperature of 25°C or 37°C.

## Mouse immunization

Female mice were divided into six groups of ten individuals. Three control groups were inoculated with saline plus adjuvant. Three experimental groups immunized with different formulations of rSm14 (25 µg/dose/mouse) with the following adjuvants: (i) MPLA-SM VacciGrade adjuvant (20 µg/dose/mouse; InvivoGen, USA); (ii) both MPLA-SM VacciGrade (20 µg/dose/mouse) and Alum – Alhydrogel (1mg/dose/mouse; InvivoGen, USA) combined; or (iii) Freund's adjuvant (100 µL/dose/mouse; Sigma-Aldrich). In the latter case, complete Freund's adjuvant (CFA) was used for the first inoculation/immunization and incomplete Freund's adjuvant (IFA) for the subsequent two booster doses. Each mouse received three doses of their respective control or vaccine formulations in a 15-day interval. All 200 µL/dose/mouse were subcutaneously inoculated (S1 Fig).

## Challenge infection and worm burden recovery

Fifteen days after the last booster dose, mice were challenged with 30 or 100 cercariae by percutaneous exposure to the abdominal skin for one hour. Fifty days after infection, challenged mice were perfused by the hepatic portal veins to recover adult worms [24]. The level of protection induced by the vaccine was determined by comparing the number of worms recovered from each experimental group to its respective control group, as previously described [23]. After perfusion, the liver and intestine of each mouse were removed, weighed, and digested in a 10% KOH solution at room temperature (RT). Schistosome eggs were obtained from digested tissue by centrifuging at 900 x g for 10min, and resuspending in 1mL of 0.85% saline. The number of eggs trapped in the liver and intestine was determined using light microscopy.

## Measurement of specific IgG levels

Specific anti-rSm14 IgG, IgG1, IgG2c and anti-SWAP IgG antibody levels were measured by ELISA. Following immunization, serum samples from control and vaccinated mice were collected 15 days after each immunization dose. Briefly, 96-well maxisorp microtiter plates (Nunc, USA) were coated with rSm14 (0.25 µg/mL) or SWAP (1 µg/mL) for IgG and rSm14 (1 µg/mL) for IgG1 and IgG2c detection in carbonate-bicarbonate buffer (pH 9.6) overnight at 4°C. The plates were washed with PBST (0.05% Tween-20 in phosphate-buffered saline (pH 7.2)) and blocked with 300 µL/well of 3% (IgG) or 10% (IgG1 and IgG2c) skim milk powder in PBST, for two hours RT. One hundred microliters of each serum sample, diluted 1:100 (IgG), 1:400 (IgG1) or 1:50 (IgG2c) in 1% skim milk powder PBST, were added to the plates and incubated for one hour at RT. The plates were then washed and incubated for one hour with HRP-conjugated goat anti-mouse IgG (1:10,000) or IgG1 (1:40,000) or IgG2c (1:15,000) (Southern Biotech, USA). TMB substrate (Sigma-Aldrich, USA) was then added to the plates, and the reaction was stopped with 5% sulfuric acid. Absorbance was measured at 450nm using a microplate reader (Thermo Scientific, USA). Endpoint antibody titers were determined using a pool of serum sample

from each group. The serial dilution ranged from 1:50 (IgG2c) or 1:200 (IgG and IgG1) to 1:1,638,400. The cut-off point for seropositivity was determined using the mean absorbance observed in blank wells plus two standard deviations. The endpoint titer was determined by the highest dilution where the absorbance was above the cut-off point.

## Immunophenotypic analysis

The cellular immune response was evaluated in blood samples from mice collected 15 days after each immunization. Blood samples from two mice of the same group were pooled. Therefore, five pools per group were analyzed. To analyze the phenotype of circulating T and B cell subsets, blood aliquots were incubated for 20 minutes at RT with monoclonal antibodies using the following combinations: anti-CD3-FITC (clone 145-2C11), anti-CD4-Alexa Fluor 700 (clone GK1.5), anti-CD62L-APC-eFluor 780 (clone MEL-14), anti-CD27-PE (clone LG.7F9), anti-CD127-PE-Cy5 (clone A7R34) (all from eBioscience); anti-CD8-APC (clone 53–6.7), anti-CD44-Pacific Blue (clone IM7) (both from BioLegend); and anti-CD19-PE-Cy-7 (clone 1D3) (BD Pharmingen). For myeloid populations, blood aliquots were incubated with monoclonal antibodies using the following combinations: Ly6C-FITC (clone AL-21), CD11b-PE-Cy-7 (Clone M1/70) from BD Pharmingen; MHC-PE (clone M5/114.15.2), F4/80-PerCP- Cy-5 (clone BM8), Ly6G-APC-eFluor-780 (clone 1A8-Ly6g) (all from eBioscience); and CD86-APC (clone GL-1), CD11c-Alexa Fluor 700 (clone N418) (both from BioLegend). Red blood cells were then lysed and washed with 2% inactivated fetal bovine serum in PBS. Cell suspensions were acquired using an LSRFortessa (Becton Dickinson, San Jose, CA), and the data were analyzed using the FlowJo 10.7.2 software (Tree Star, Ashland). A baseline fold change index was obtained by dividing the mean of the experimental groups by the mean of saline/adjuvant. Values above and below one indicate an increase and decrease in the frequency of a cell population after immunization, respectively.

## Cytokine analysis

Cytokine production induced by different vaccine formulations was assessed in sera samples after each immunization dose using the Cytometric Bead Array anti-mouse CBA Th1/Th2/Th17 Kit (BD Pharmingen, USA), according to the manufacturer's protocol. The detection limits for each cytokine, as reported by the manufacturer, were as follows: IL-2 (0.1 pg/mL), IL-4 (0.03 pg/mL), IL-6 (1.4 pg/mL), IFN-γ (0.5 pg/mL), TNF (0.9 pg/mL), IL-17 (0.8 pg/mL), and IL-10 (16.8 pg/mL). Data were acquired using a FACSVerse flow cytometer (BD, USA), and analyzed by the FCAP Array 3.0 Software (BD, USA).

## Statistical analysis

GraphPad Prism 10.0 (Graph-Pad Software, USA) was used to perform statistical analyses. Initially, the data were submitted to the Shapiro-Wilk normality test. Two-way ANOVA test was used to calculate differences among groups and over time for immunophenotyping, production of IgG and cytokines. The number of worms recovered and the number of eggs in the liver and intestine were analyzed using an unpaired Student's t-test, comparing the vaccinated group with the respective saline control group.

## Results

### Expression and purification of the recombinant Sm14 protein

The synthetic *sm14* gene, optimized for expression in prokaryotic systems, and the protein sequence are represented in Fig 1A and 1B, respectively. The replacement of a cysteine residue at position 62 for a valine is highlighted. A prominent protein band of approximately 14 kDa was visualized in the bacterial lysates in a 15% SDS-PAGE (Fig 1C, lanes 2 and 3), compared with non IPTG induced bacterial lysate (Fig 1C, lane 1). A high purity fraction of rSm14 was identified after nickel-affinity purification (Fig 1C, lane 4). A monoclonal 6x-His-tag antibody recognized two bands, each with a molecular

**A**

## Synthetic gene

```
1    GAATTCATGAGCAGCTTTCTGGGCAAATGGAAACTGAGCGAG
43   AGCCACAACTTCGATGCGGTTATGAGCAAACTGGGTGTTAGC
85   TGGGCGACCCGTCAGATCGGTAACACCGTGACCCCGACCGTT
127  ACCTTCACGATGGACGGCGATAAAATGACCATGCTGACCGAG
169  AGCACCTTCAAAAACCTGAGCGTGACCTTCAAGTTTGGCGAG
211  GAATTTGACGAAAAGACCAGCGATGGCCGTAACGTTAAGAGC
253  GTGGTTGAGAAGAACAGCGAAAGCAAGCTGACCCAGACCCAA
295  GTGGACCCGAAAAACACCACCGTGATTGTTCGTGAGGTTGAC
337  GGTGATACCATGAAGACCACCGTGACCGTGGGTGATGTGACC
379  GCGATTCGTAACTACAAACGCCTGAGCCTCGAG
```

**B**

## *Schistosoma mansoni* rSm14

```
1    MSSFLGKWKLSESHNFDAVMSKLGVSWATRQIGNTVTPTVTF
43   TMDGDKMTMLTESTFKNLSVTFKFGEEFDEKTSDGRNVKSVV
85   EKNSESKLTQTQVDPKNTTVIVREVDGDTMKTTVTVGDVTAI
127  RNYKRLS
```

**C**

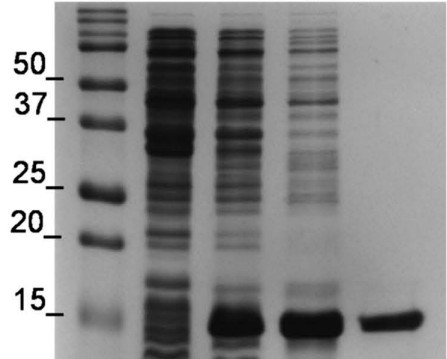

**D**

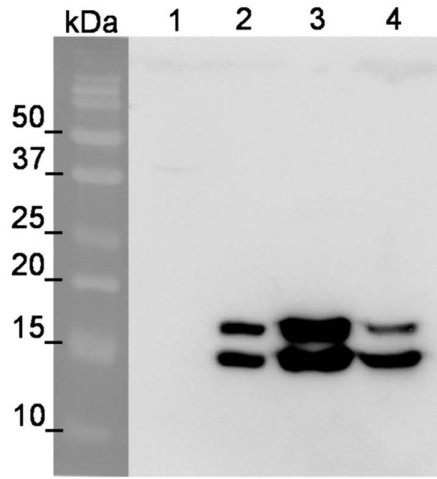

**E**

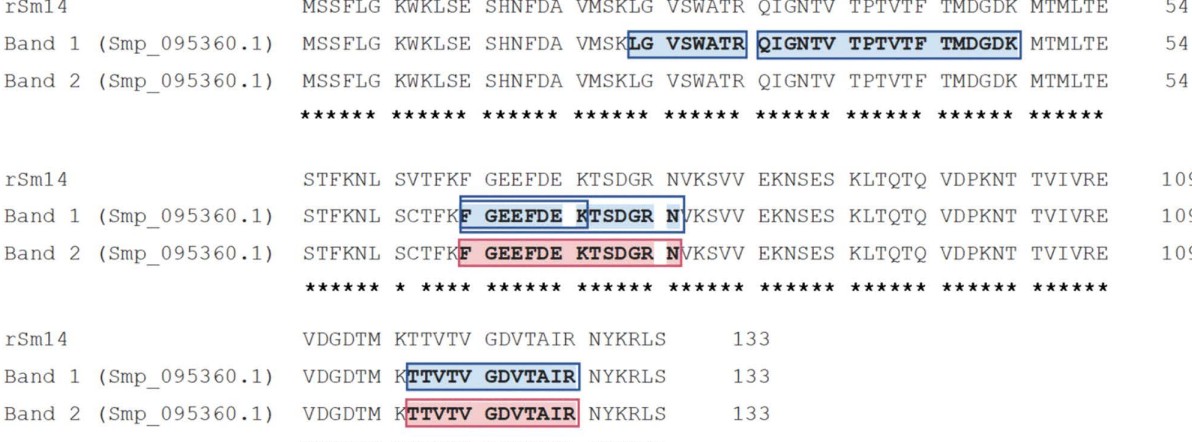

**Fig 1. Expression and purification of the recombinant *Schistosoma mansoni* Sm14.** DNA sequence of the synthetic gene corresponding to the coding region of the Sm14 protein. The restriction sites for the enzymes *EcoRI* and *XhoI* are highlighted in blue at the 5′ and 3′ ends, respectively, and the initiation codon ATG is denoted in pink **(A)**. Amino acid sequence of the recombinant *Schistosoma mansoni* Sm14 protein **(B)**. The substitution of

cysteine for valine at position 62 is highlighted in bold in both the nucleotide and amino acid sequences. 15% SDS-PAGE **(C)** and Western blot **(D)** using a monoclonal 6x-His-tag antibody for the analysis of rSm14 expression and purification: non-IPTG induced (lane 1); 1mM IPTG-induced protein expression in bacterial culture (lane 2); bacterial lysate under denaturing conditions (lane 3); and purified rSm14 fraction (lane 4). Molecular weight markers used were Precision Plus Protein™ Dual Color Standards Bio-Rad. Alignment of the rSm14 amino acid sequence with the peptide sequences detected by mass spectrometry analysis **(E)**. For the first band, five unique peptides were detected (covered amino acids sequence highlighted in blue). For the second band, two unique peptides were detected (covered amino acids sequence highlighted in pink).

mass of approximately 14 kDa, suggesting that the purified protein corresponds to rSm14 (Fig 1D). Peptides from the protein Smp_095360.1, which corresponds to the 14 kDa fatty acid-binding protein Sm14, were detected by mass spectrometry in both protein bands. Five unique peptides from Sm14 were detected in band 1, resulting in a sequence coverage of 38%, and two unique peptides and a coverage of 19% for band 2 (Fig 1E, S1 Table). The analysis of the trypsin proteolysis assay indicates a rapid degradation of heat-denatured rSm14, with approximately 50% loss of intact protein within 40 minutes at 25°C, whereas only a minor degradation of the rSm14 was observed in the SDS-PAGE, with more than 80% of protein remaining intact after 60 minutes of digestion (S2 Fig).

### Different vaccine formulations containing rSm14 induce distinct effector and memory profile of CD4$^+$ and CD8$^+$ T cells

Immunophenotypic analysis was performed using blood from mice 15 days after each immunization dose. The profile of circulating T cells was determined using the analysis strategy shown in panel 2A (Fig 2). Inoculation with rSm14/MPLA induced increased proportions of CD4$^+$ T cells (CD3$^+$CD4$^+$) after the first and second immunization doses compared to the saline group (Fig 2B,2C). We also observed a slight increase in the proportion of effector (CD44$^+$CD62L$^-$CD127$^-$) cells in this group after the first dose and significant increase after the second dose (Fig 2B,2D). All formulations induced higher frequencies of effector cells after the third dose compared to the previous one, except for the rSm14/MPLA formulation, in which the increase was only observed compared to the second dose (Fig 2D). An expansion of effector memory (EM, CD62L$^-$CD127$^+$CD44$^+$) CD4$^+$ T cells was observed in mice immunized with three doses of rSm14/MPLA/Alum (Fig 2B,2E). A decrease in EM cells was observed in the rSm14/Freund's group after the second and third doses and in the rSm14/MPLA group after the third dose compared to the first one (Fig 2E). Additionally, higher frequencies of this population were observed in rSm14/MPLA/Alum group after the third compared to the second dose (Fig 2E). Increased frequencies of central memory (CM, CD44$^+$CD62L$^+$CD127$^+$) (Fig 2B,2F) CD4$^+$ T cells were observed only after the third dose of rSm14/MPLA. The vaccine booster induced an increase in the frequency of CM cells in the rSm14/MPLA group after the second and third relative to the first dose. For the rSm14/Freund's group, no significant difference was observed in the frequency of EM and CM cells compared to the saline group (Fig 2B,2E). Furthermore, we observed a reduction in the cells after the third compared to the second dose in the rSm14/Freund's group and to the first and second doses in the rSm14/MPLA/Alum group (Fig 2F).

    rSm14/MPLA formulation induced a reduction in the proportion of CD8$^+$ T cells after all immunization doses (Fig 2G,2H). Additionally, a decrease in the proportions of effector CD8$^+$ T cells was observed after the second dose of rSm14/Freund's formulation (Fig 2G,2I). In the rSm14/Freund and rSm14/MPLA/Alum groups the third dose induced increased proportions of effector CD8$^+$ T cells than the previous doses (Fig 2I). All vaccine formulations containing rSm14 induced a decrease in the frequency of EM CD8$^+$ T cells after the second dose (Fig 2G,2J). We found an increase in the proportion of EM cells in the rSm14/MPLA/Alum group after third doses compared to the first and second doses (Fig 2G,2J). Both rSm14/Freund's and rSm14/MPLA/Alum formulations induced an increase in the proportions of CM cells after the second dose (Fig 2G,2K). Finally, the third dose of immunization triggered a reduction in the proportion of CM cells in rSm14/MPLA/Alum group, compared to the second dose (Fig 2K).

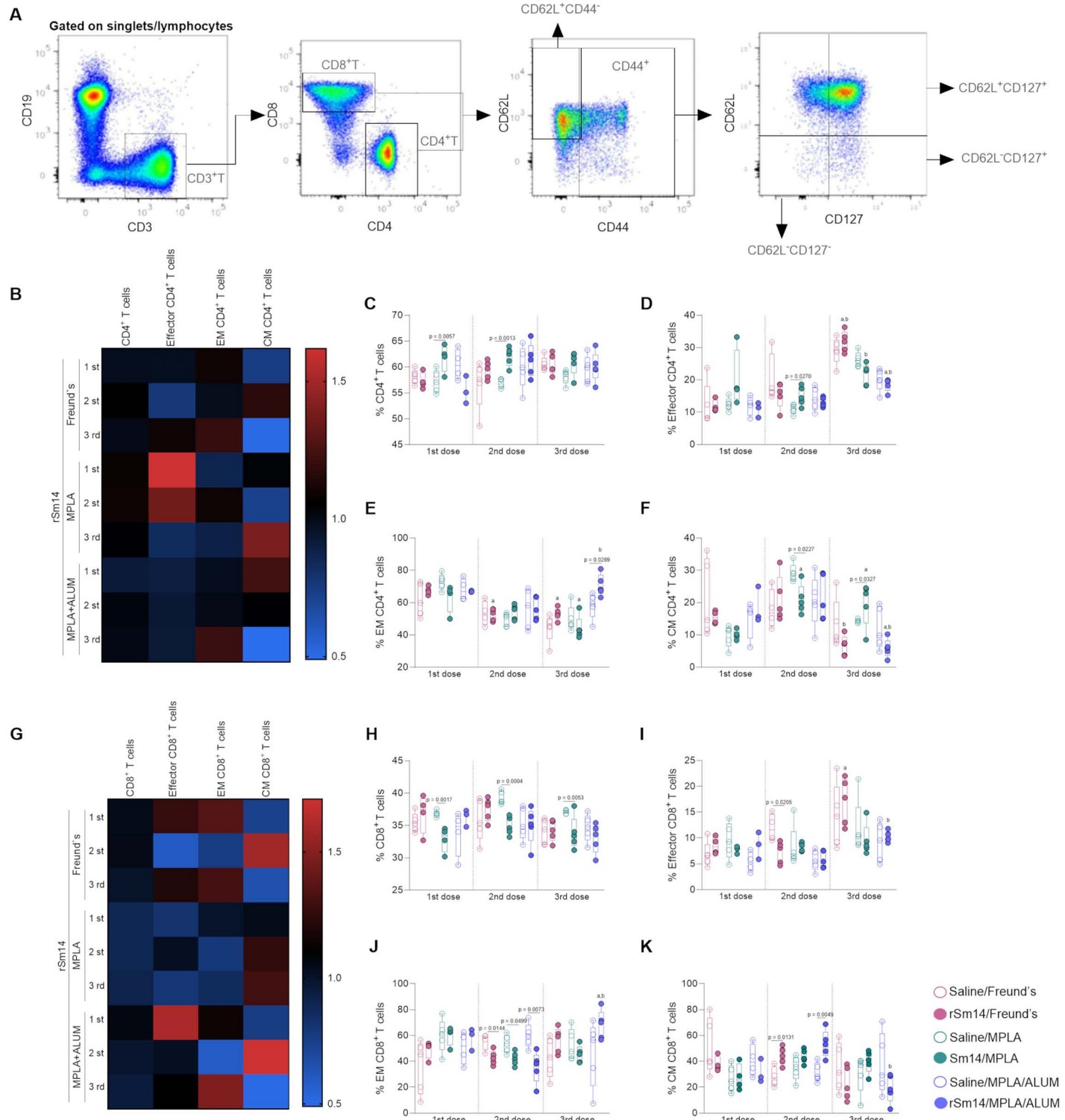

**Fig 2. Immunophenotypic profile of circulating CD4⁺ T and CD8⁺ T cells from rSm14-immunized mice.** Representative flow cytometry dot plots **(A)**. Heatmap depicts the fold change relative to controls for the following populations: total CD4⁺ T and effector, effector memory (EM) and central memory (CM) CD4⁺ T cells. Values above 1 indicate an increase and values below 1 indicate a reduction in the frequency of a cell population after immunization **(B)**. Box with whiskers graphs showing the distributions of CD4⁺ T cell populations **(C-F)**. Heatmap depicts the fold change relative to

controls for the following populations: total CD8$^+$ T, effector, effector memory (EM), and central memory (CM) CD8$^+$ T cells **(G)**. Box with whiskers graphs showing the distributions of CD8$^+$ T cell populations **(H-K)**. Each of the five symbols represents a pool of cells from two animals in the group (3-5 pools per group)**(C-F, H-K)**. Differences were determined by two-way ANOVA followed by Tukey's multiple comparisons test. The letters "a" and "b" denote statistically significant differences in relation to the first and second immunization doses, respectively.

## rSm14/Freund vaccine formulation induces a stronger humoral immune response and promotes an expansion in memory B cells

Mice vaccinated with rSm14/Freund's showed higher B cell frequencies (CD19$^+$) after the first and second immunization doses (Fig 3B,2C). We also observed a reduction in the proportion of B cells in all vaccine formulations after the second and third doses compared to the first dose, except for the rSm14/MPLA/Alum group, in which a decrease was only observed after the third dose. The third dose of all vaccine formulations promoted an increase in the frequency of memory B cells compared to the previous ones (Fig 3B,2D). Moreover, only rSm14/Freund's formulation induced an increase in the frequencies of memory (CD27$^+$) B cells compared to its control group (Fig 3B,2D).

All vaccine formulations induced significant production of anti-rSm14 IgG (Fig 3E) and IgG1 (Fig 3F) antibodies compared to their respective control groups, 15 days after the second and third immunization doses. However, for IgG2c, this increase was only observed in the rSm14/Freund's group (Fig 3G). After the second and third doses, IgG (Fig 3E) and IgG2c (Fig 3G) levels were higher in the rSm14/Freund's than in the other immunized groups. The rSm14/Freund's and rSm14/MPLA/Alum groups presented higher levels of IgG1 compared to the rSm14/MPLA group after the second and third doses (Fig 3F). Each booster induced an increase in the levels of IgG (Fig 3E) and IgG1 (Fig 3F) in all groups, except rSm14/MPLA, in which the increase was only observed after the third dose. IgG2c levels also increased after the second and third boosters in rSm14/Freund's group compared to the previous doses (Fig 3G). Antibodies produced against rSm14 also recognized the soluble worm antigen preparation (SWAP), which contains the protein native form (S3 Fig).

Furthermore, we evaluated endpoint antibody titers in all groups (Table 1, S4–S6 Fig). The endpoint titer of anti-rSm14 IgG in mice immunized with rSm14/Freund's was 8- and 16-fold higher than rSm14/MPLA/Alum, and 64- and 32-fold higher than rSm14/MPLA, after the second and third doses, respectively. Moreover, the rSm14/MPLA/Alum group showed IgG titers 8- and 2-fold higher than the Sm14/MPLA group, after the second and third doses, respectively (Table 1, S4 Fig). Differences were also noted in the rSm14-specific IgG1 endpoint titer (Table 1, S5 Fig). Mice immunized with rSm14/MPLA/Alum produced 2-fold higher titers than rSm14/Freund's after the first dose, but equal titers were observed after the second and third doses. In addition, after the first dose, animals immunized with rSm14/Freund's and rSm14/MPLA/Alum presented, respectively, antibody titers 4 and 8-fold higher than rSm14/MPLA group. A 16-fold increase in IgG1 titers compared to those observed in rSm14/MPLA was detected in the other immunization groups after the second and third doses (Table 1, S5 Fig). Regarding IgG2c (Table 1, S6 Fig), rSm14/Freund's induced titers 4- and 64-fold higher than the other formulations after the first and second doses, respectively. After the third dose, the rSm14/Freund's group produced 128- and 8-fold higher IgG2c titers than the rSm14/MPLA and rSm14/MPLA/Alum groups, respectively. The addition of Alum to the formulation rSm14/MPLA induced a 16-fold increase in this antibody titers (Table 1, S6 Fig).

## rSm14 vaccine formulations induce increase in the frequencies of neutrophils and monocytes

Neutrophils and monocytes were assessed in the blood from mice 15 days after each immunization dose using flow cytometry, as depicted in panel A (Fig 4A). The proportion of neutrophils (F4/80$^-$Ly6G$^+$) (Fig 4B,4D) was higher in the rSm14/MPLA/Alum group compared to its control group after the first and third immunization doses. rSm14/Freund's and rSm14/MPLA/Alum formulations induced a significant increase in the proportions of monocytes (F4/80$^+$Ly6G$^-$) after the first and third doses, respectively, when compared to their respective control groups (Fig 4C,4D).

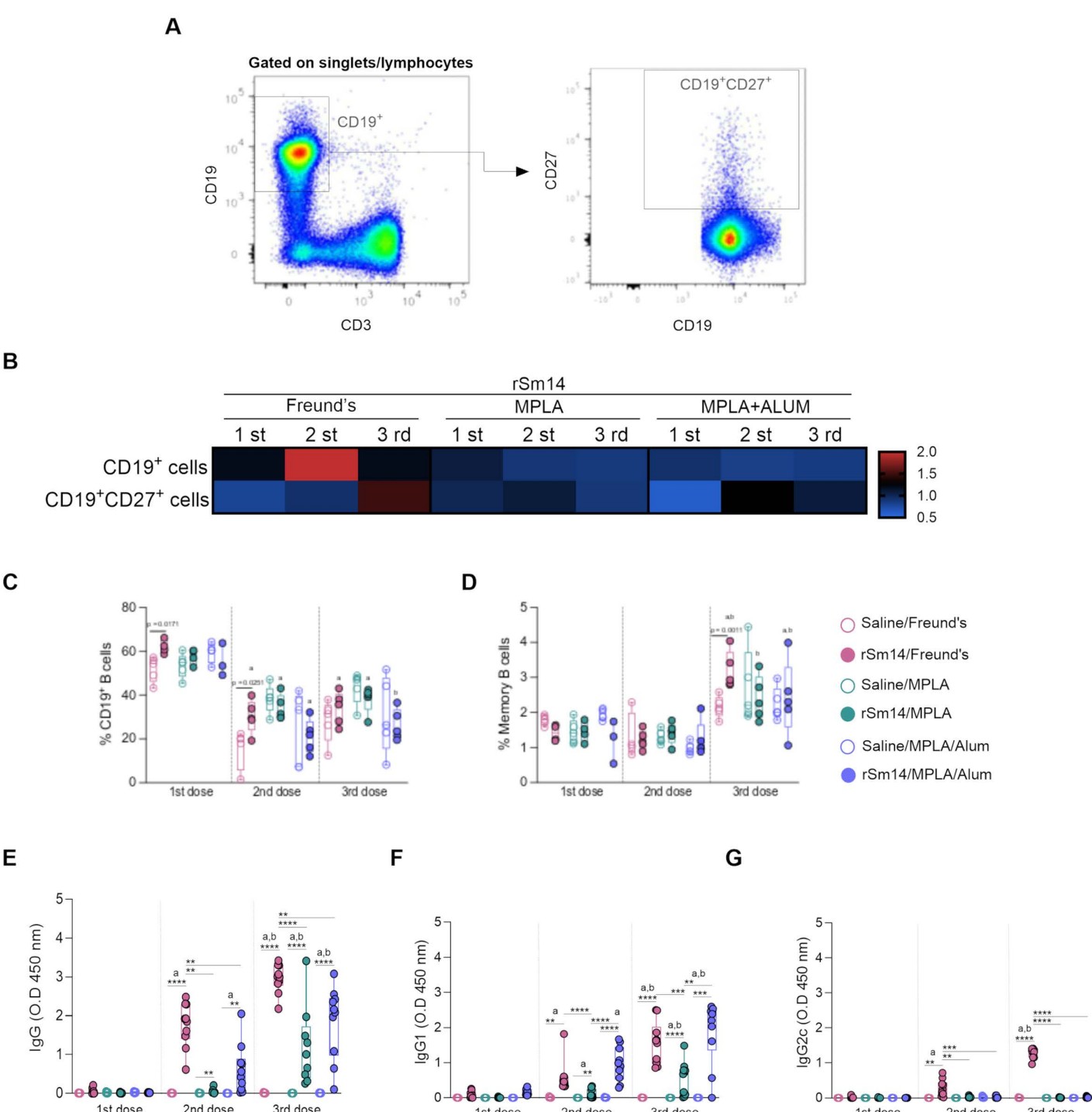

**Fig 3. Immunophenotypic profile of circulating B cells and antibody production in rSm14-immunized mice.** Representative flow cytometry dot plots showing the frequencies of B cell populations **(A)**. Heatmap depicts the fold change relative to controls for the following populations: CD19+ B cells and memory B cells. Values above 1 indicate increase and values below 1 indicate reduction in the frequency of a cell population after immunization **(B)**. Box with whiskers graphs showing the distributions of total CD19+ B cells **(C)** memory B cells **(D)** and levels of IgG, IgG1 and IgG2c **(E-G)**. Each of the five symbols represents a pool of cells from two animals in the group (3-5 pools per group) **(C,D)** or represents an animal (5-10 mice per group) **(E-G)**. Differences were determined by two-way ANOVA followed by Tukey's multiple comparisons test. The letters "a" and "b" denote statistically significant differences in relation to the first and second immunization doses, respectively.

**Table 1. Endpoint titers of specific IgG antibodies against rSm14 in immunized mice.**

| | 1st dose | | | 2nd dose | | | 3rd dose | | |
|---|---|---|---|---|---|---|---|---|---|
| | IgG | IgG1 | IgG2c | IgG | IgG1 | IgG2c | IgG | IgG1 | IgG2c |
| rSm14/Freund's | 400 | 1,600 | 400 | 25,600 | 51,200 | 6,400 | 204,800 | 409,600 | 102,400 |
| rSm14/MPLA | <400 | 400 | <50 | 400 | 3,200 | 100 | 6,400 | 25,600 | 800 |
| rSm14/MPLA/Alum | <400 | 3,200 | <50 | 3,200 | 51,200 | 100 | 12,800 | 409,600 | 12,800 |

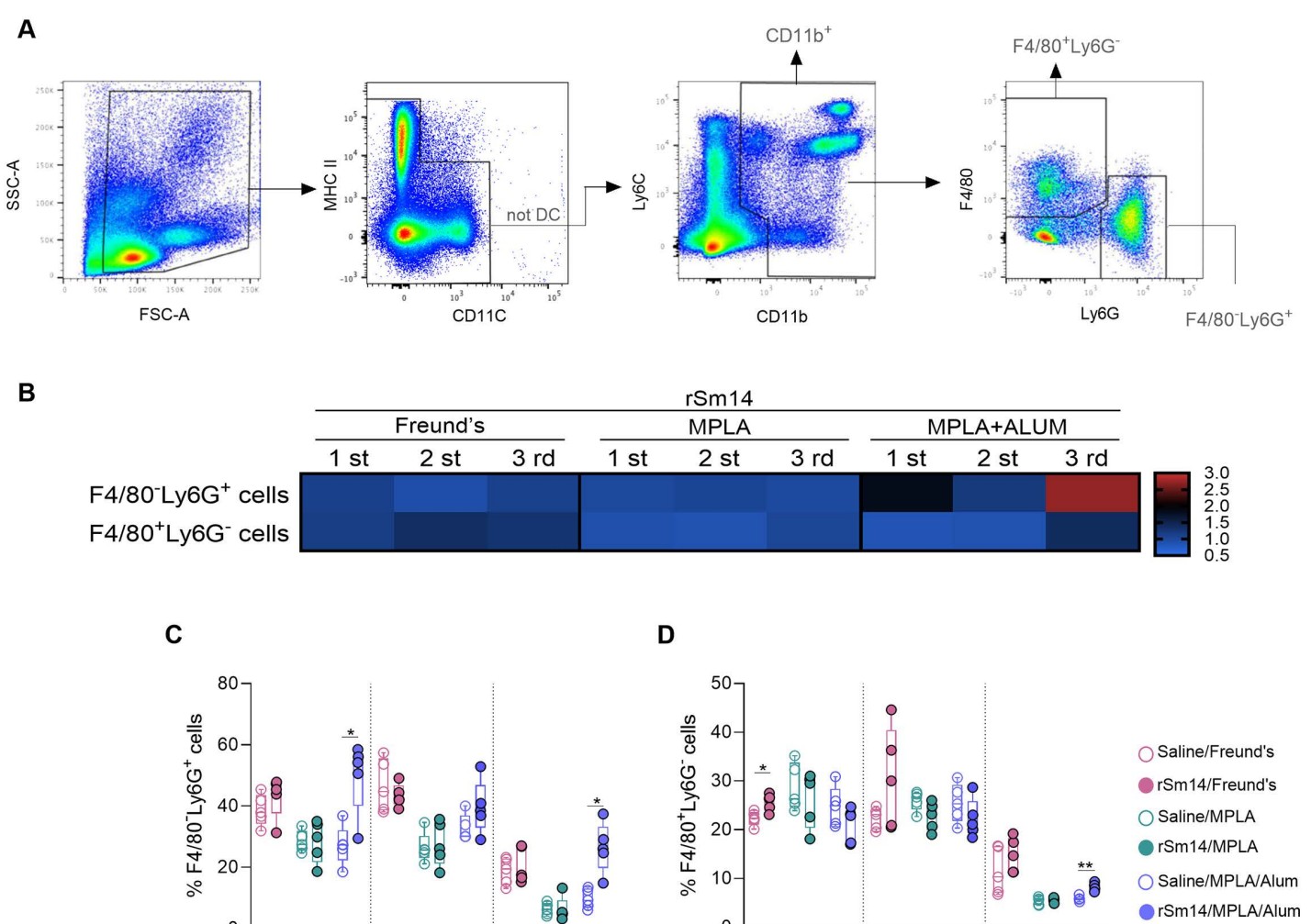

**Fig 4. Immunophenotypic profile of circulating neutrophil and monocyte from rSm14-immunized mice.** Representative flow cytometry dot plots (A). Box with whiskers graphs showing the distributions of neutrophils (F4/80⁻Ly6G⁺) (B) and monocytes (F4/80⁺Ly6G⁻). Heatmap depicts the fold change relative to controls for F4/80⁻Ly6G⁺ and F4/80⁺Ly6G⁻ (C). Values above 1 indicate increase and values below 1 indicate reduction in the frequency of a cell population after immunization. Each of the five symbols represents a pool of cells from two animals in the group (5 pools per group) (B,C). Differences were determined by two-way ANOVA followed by Tukey's multiple comparisons test.

## Sm14/Freund's formulation induced a higher production of circulating TNF, IFN-γ and IL-6

Among vaccine formulations, rSm14/Freund's induced the highest levels of TNF regardless the immunization dose and of IFN-γ and IL-6 after the second and third doses (Fig 5A–5C). Addition of Alum to the rSm14/MPLA formulation induces an increase in the levels of IL-6 after the third dose (Fig 5C). After the third dose, the rSm14/Freund's group showed significantly higher production of TNF and IFN-γ compared to its control group (Fig 5A,5B). After the second and third doses, there was a significant increase in IL-6 production in the rSm14/Freund's and rSm14/MPLA/Alum groups, respectively, compared to their control groups (Fig 5C). Higher levels of IFN-γ and IL-6 were observed after the third dose of rSm14/Freund's immunization compared to the first dose, and the highest levels of TNF were observed after the third rSm14/Freund's dose (Fig 5A–5C). The cytokines IL-2, IL-4, IL-10, and IL-17 had values below the detection limit.

## Vaccine formulations containing rSm14 failed to induce protection against challenge infection

Given that immunization with the different rSm14-containing vaccine formulations showed an immunostimulatory effect, we evaluated their ability to protect mice against schistosome infection. Following a challenge with 100 cercariae, none of the tested formulations, regardless of the adjuvant used, led to a significant reduction in parasite burden. To evaluate whether this lack of protection was due to an excessively high number of cercariae in the challenge infection, we performed an additional trial using a lower cercarial burden (30 cercariae). Nevertheless, no protection was observed in this setting either. Additionally, no significant reduction in the number of eggs per gram of liver and intestine was observed (Fig 6).

## Discussion

Adjuvants are important components of vaccine formulations and are essential for enhancing and directing adaptive immune responses to target vaccine antigens, mediated by B and T lymphocytes [10]. In this study, we assessed whether different adjuvant formulations combined with the rSm14 antigen, which is currently in phase 2b clinical trials, were able to induce an immunostimulatory effect (ClinicalTrials.gov ID: NCT03799510).

The success of vaccines often depends on the induction of a strong and persistent memory response [25]. However, the immunostimulatory effects induced by *Schistosoma* antigens formulated with adjuvants, especially the alterations in memory compartments, have been poorly documented. In our study we observed an increase in the frequency of central memory CD4+ T cells in mice immunized with rSm14/MPLA formulation. In agreement with this finding, Mathias and coworkers [26] observed a significant increase in the frequency of central memory CD4+ T cells in mice immunized with *Leishmania braziliensis* antigen plus saponin and MPLA. Moreover, our results also show that mice immunized with rSm14/MPLA/Alum presented a higher proportion of effector memory CD4+ T cells. In another model, *Leptospira* antigen

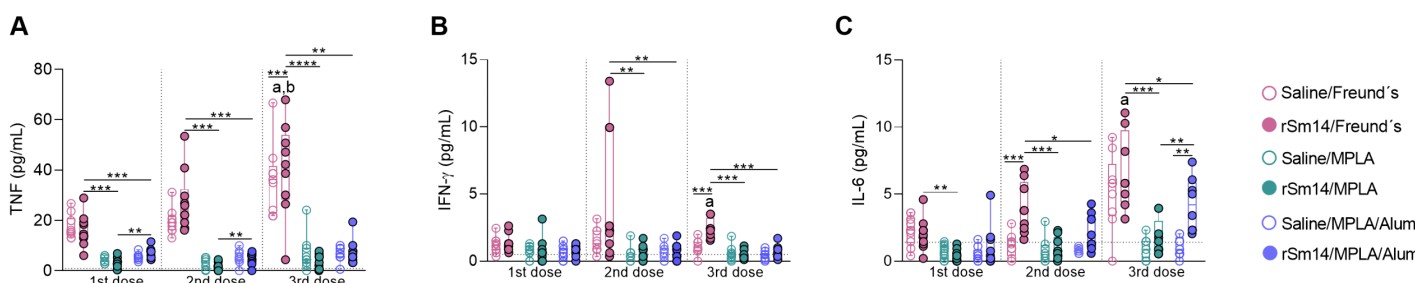

**Fig 5. Cytokine profile induced in mice immunized with rSm14.** Levels of TNF **(A)**, IFN-γ **(B)** and IL-6 were measured in serum using the CBA Th1/Th2/Th17 kit. Box with whiskers graphs represent the distributions of the circulating cytokines in different samples. Each symbol represents an animal (7-10 mice per group). Differences were determined by two-way ANOVA followed by Tukey's multiple comparisons test. The letters "a" and "b" denote statistically significant differences in relation to the first and second immunization doses, respectively.

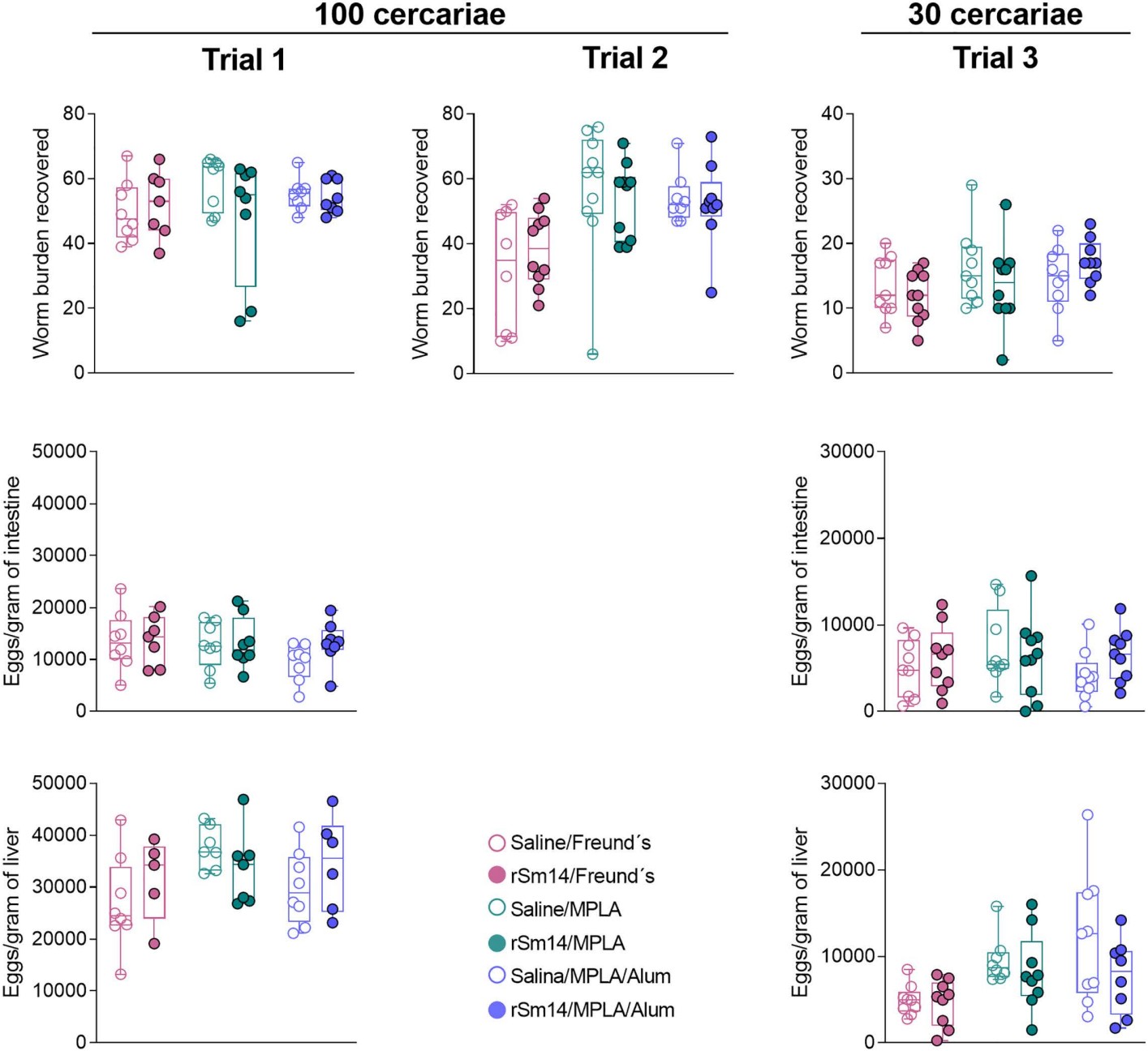

**Fig 6. Worm burden recovered, and number of eggs trapped in the liver and in the intestine of immunized mice.** Box-and-whisker plots show the distribution of worm burden recovered **(A)**, eggs retained in the intestine **(B)**, and eggs retained in the liver **(C)** fifty days after challenge infection with 100 cercariae (Trials 1 and 2) or 30 cercariae (Trial 3) in mice immunized with three doses of different rSm14-containing formulations. Each symbol represents an animal (5-10 mice per group). Statistically significant differences between groups were assessed using Student's t-test.

plus AS04 (MPLA/Alum) induced significantly higher levels of central and effector memory CD4$^+$ T cells [25]. For CD8$^+$ T cell subpopulations, rSm14/Freund's and rSm14/MPLA/Alum formulations induced an expansion of the central memory compartment. A previous study using an influenza vaccine formulation showed that the association of Alum and MPLA provides the signals necessary for the generation of a long-term memory CD8$^+$ T cell subset [27].

The ability of B cells to differentiate into memory cells is important for the success of most vaccines [28]. Our results showed that the rSm14/Freund's formulation led to an increase in the proportion of B cells and memory B cells. Bernardes et al. (2019) [23] when immunizing mice with a recombinant protein from *S. mansoni*, rSm16, in association with Freund's adjuvant also observed an increase in the frequency of memory B cells.

B cells and antibodies have been implicated in the protective effector mechanisms of anti-schistosomiasis vaccines [29–31]. Several mechanisms have been proposed to explain how antibodies induced by vaccine antigens contribute to the elimination of *Schistosoma mansoni*, including antibody-dependent cellular cytotoxicity (ADCC) and inhibition of antigen function [30,32,33]. In the case of Sm14, its predicted role in host lipid uptake and surface localization suggests that both ADCC and the inhibition of lipid acquisition could play a role in parasite killing. Although, the precise mechanisms through which anti-Sm14 antibodies mediate parasite death remain to be fully elucidated, immunoepidemiologic studies in humans have shown that the production of anti-Sm14 IgG1 and IgG3 is associated with resistance to infection and reinfection [34].

All vaccine formulations induced significant production of Sm14-specific IgG antibodies, although a more robust production was observed in the rSm14/Freund's group. In addition, different humoral profile was observed in response to vaccine formulation, with rSm14/MPLA/Alum and rSm14/MPLA inducing mainly IgG1 production while rSm14/Freund's formulation inducing both IgG1 and IgG2c production. In agreement with these findings, a study demonstrated that an HPV vaccine formulated with MPLA/Alum significantly increased the production of specific IgG1 antibodies [35]. In contrast, the use of Freund`s adjuvant in association with *S. mansoni* antigens increases the production of IgG2a/IgG2c antibodies [23,36,37].

Previous studies using recombinant forms of the Sm14 protein in association with Freund's adjuvant demonstrated protection against *S. mansoni* infection [12,13] associated with increased production of IgG2a, which correlates with C57BL/6 IgG2c and human IgG1, and was dependent on IFN-γ and TNF [13]. In a Phase 1 clinical trial, rSm14 protein formulated with glucopyranosyl lipid A (GLA-SE) induced a significant increase in the production of IgG antibodies and subclasses, as well as cytokines such as IFN-γ after immunizations. Although this study does not allow a direct association of resistance and immunogenicity, the authors suggested that the production of these factors could be associated with resistance to infection [16]. In our study, only rSm14/Freund's induced an increase in IFN-γ and TNF levels and in the production of higher titters of specific IgG2c antibodies. Indeed, it is reported that Freund's Complete adjuvant leads to a Th1 response [38], while Alum induces a Th2 immune profile [8].

Adjuvants have a role in the innate immune response, expanding the myeloid population [39]. Our findings showed that therSm14/Freund's and rSm14/MPLA/Alum formulations induced an increase in the proportion of monocytes. These cells respond to a wide array of signaling molecules, including IFN-γ and TNF, which induce activation of proinflammatory genes, chemokines, additional cytokines and nitric oxide (NO), and this environment induces differentiation of inflammatory macrophages [40]. Nitric oxide produced by IFN-γ-activated macrophages plays an important role in *S. mansoni* killing. Mice deficient in inducible nitric oxide synthase (iNOS) have a reduced ability to eliminate *S. mansoni* in vivo, reinforcing the importance of NO for protection [41]. Likewise, an increase in the proportion of neutrophils along with increased production of IL-6 in mice immunized with rSm14/MPLA/Alum was observed. Indeed, studies in vitro suggested a role for neutrophils in *Schistosoma* killing [42,43]. IL-6 enhances signal transducer and activator of transcription 3 (STAT3) activation and neutrophil recruitment [44] transition, through a shift of chemokine production [45]. In addition, Alum stimulates IL-1β secretion at the injection site, which also induces neutrophil influx *in vivo* [46].

To evaluate if the immune response in mice correlates with protective efficacy, the animals were infected with 30 or 100 cercariae, 15 days after the last dose of immunization. However, although all vaccine formulations induced both cellular and humoral responses, we did not find protection against *S. mansoni*, independent of the parasite load used in the infection. The protection induced by Sm14 is dependent on its tridimensional structure [47], thus, differences in its sequence and folding can compromise the ability to induce protection. Tendler et al. (1996) [12] and Fonseca et al. (2005)

[13] demonstrated that rSm14 induce protection using recombinant forms of Sm14 in fusion with the product of gene 10 of bacteriophage T7 and the maltose-binding protein, respectively. Here, we produced a recombinant protein fused to a 6×His tag. Our data demonstrate that the recombinant Sm14 used in the vaccine formulations is less susceptible to proteolysis than its heat-denatured form, suggesting stable protein folding. Additionally, antibodies raised against rSm14 can recognize the native Sm14 in SWAP, indicating that rSm14 retains epitopes from its native conformation. However, the protective functional activity of these antibodies against the parasite remains to be demonstrated.

In this study, we describe the profile of the immune response triggered by the vaccine candidate rSm14 against schistosomiasis in association with different adjuvants. Our data showed that each vaccine formulation induced distinct profiles of cellular and humoral response. While vaccination with formulations containing MPLA induced an expansion in the memory T cell compartment, Freund`s adjuvant mainly promote the expansion of memory B cells and the production of soluble factors as IgG2c, IFN-γ and TNF (Fig 7; S2 Table). The lack of protection even in an immune environment associated with parasite killing indicates that the use of the molecules evaluated in this work should be carefully considered as a correlate of protection

## Supporting information

**S1 Fig. Experimental procedures.** For the production of rSm14, a synthetic gene for expression in *Escherichia coli* was constructed. Gene expression was induced with IPTG. After bacterial lysis under denaturing conditions, rSm14 was purified by nickel affinity chromatography. Purified rSm14 was then dialyzed against phosphate-buffered saline. The identity of the purified protein was confirmed by mass spectrometry. Mice received three doses of each vaccine in 15-day interval regimen (rSm14 + Freund, rSm14 + MPLA, rSm14 + MPLA+Alum and their respective saline/adjuvant control groups). Animals were challenged by percutaneous infection with 30 or 100 cercariae 15 days after the last immunization. Fifty days after infection, adult worms were recovered from the portal and mesenteric veins and the number of eggs present in liver and intestine was determined. Fifteen days after the first, second and third doses of immunization, the phenotype of CD4+ T cells, CD8+ T cells, B cells and myeloid population for activation and memory markers were evaluated in blood samples. Levels of rSm14-specific IgG, IgG1, IgG2c, SWAP-specific IgG antibodies and circulating cytokines were measured after each immunization dose. Created in BioRender. Vacinas, C. (2025) https://BioRender.com/yn33kjp.
(TIF)

**S2 Fig. Decay of intact rSm14 protein under trypsin proteolysis.** Heat-denatured rSm14 **(A)** and rSm14 **(B)** were subjected to trypsin digestion at 25°C for 0 (T0), 1 (T1), 2 (T2), 5 (T5), 10 (T10), 20 (T20), 40 (T40), and 60 (T60) minutes. The degradation profiles were analyzed by 15% SDS-PAGE and visualized with Coomassie Brilliant Blue G-250 **(A–B)**. Molecular weight markers (MW) used were Precision Plus Protein™ Dual Color Standards (Bio-Rad). Densitometric analysis of the intact rSm14 band over time was performed using ImageJ **(C)**. T0 values were used for normalization, and results are expressed as the percentage of intact protein with the Exponential trendline.
(TIF)

**S3 Fig. Recognition of SWAP by sera from mice immunized with rSm14.** Sera from mice immunized with rSm14 were obtained 15 days after each vaccine dose to evaluate the recognition of SWAP. Box and whiskers represent the distribution of IgG antibody levels in different sera. Each symbol represents an animal (9–10 mice per group). Differences were determined by two-way ANOVA followed by Tukey's multiple comparisons test. The letters "a" and "b" denote statistically significant differences in relation to the first and second immunization doses, respectively.
(TIF)

**S4 Fig. rSm14-specific IgG antibody titer endpoints in immunized mice serum.** Pools of sera from mice immunized with rSm14 + Freund's (A), rSm14 + MPLA (B) or rSm14 + MPLA+Alum (C) obtained 15 days after the first, second and

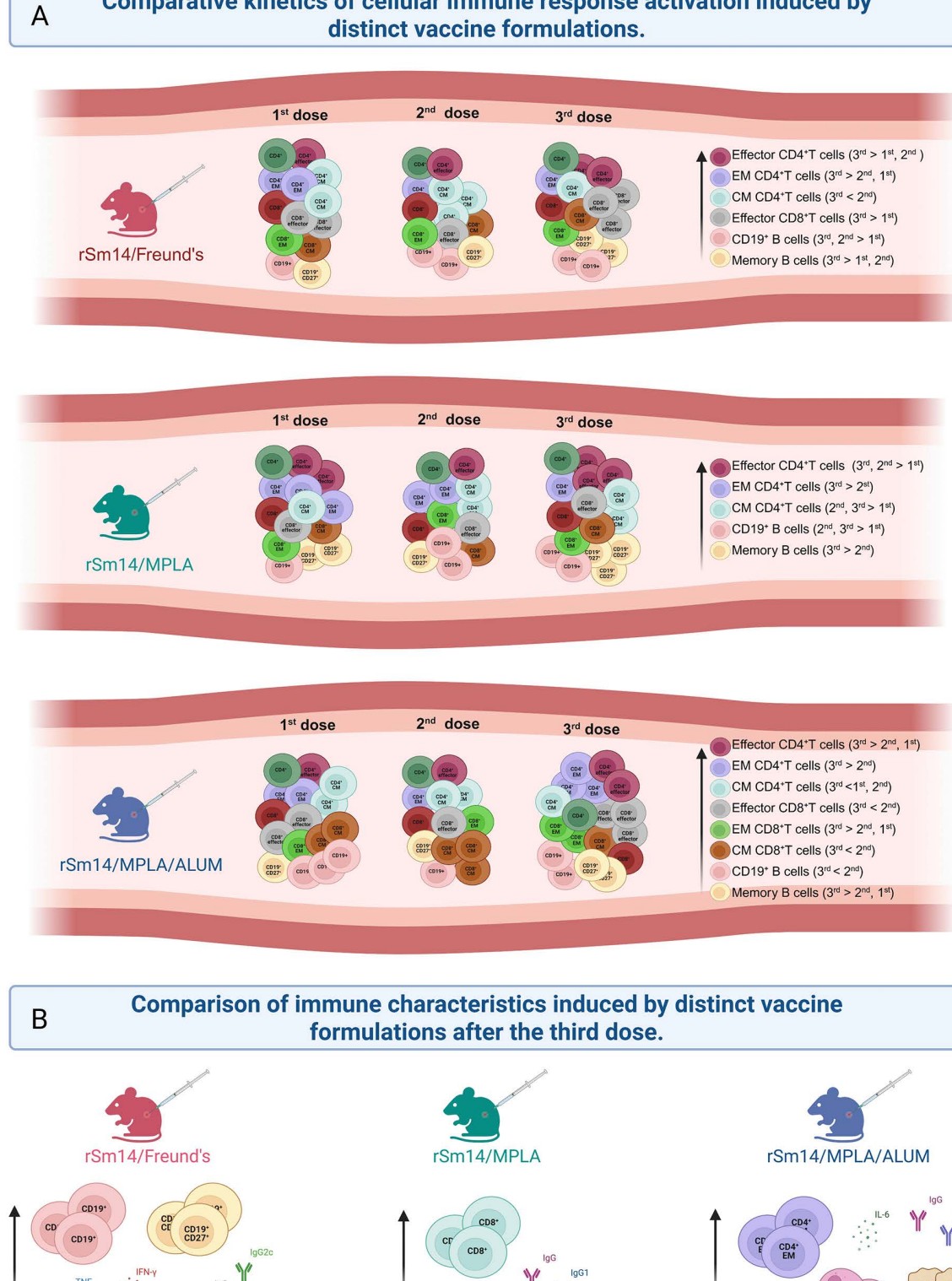

**Fig 7. Overview of the immune response triggered by rSm14 vaccine formulations.** Mice received three doses of each vaccine in a 15-day interval regimen (rSm14 + Freund, rSm14 + MPLA, rSm14 + MPLA+Alum and their respective saline/adjuvant control groups). Fifteen days after the first, second

and third doses of immunization, the proportion of F4/80$^-$Ly6G$^+$ (neutrophils), F4/80$^+$Ly6G$^-$ (monocytes), activated and memory CD4$^+$ T cells, CD8$^+$ T cells, B cells, IgG subclasses, and cytokines were evaluated in blood samples. The kinetics of cellular immune response activation induced by different vaccine formulations is shown in panel A and the comparison of immune characteristics induced by different vaccine formulations after the vaccination scheme (third dose) is shown in panel **B.** EM: effector memory; CM: central memory.

third immunization doses were serially diluted, beginning at 1:200, and used in an ELISA assay. The threshold was calculated using the mean absorbance value of the blank wells plus two standard deviations. The arrows indicate antibody titer endpoints observed after immunization.
(TIF)

**S5 Fig. rSm14-specific IgG1 antibody titer endpoints in immunized mice serum.** Pools of sera from mice immunized with rSm14 + Freund's (A), rSm14 + MPLA (B) or rSm14 + MPLA+Alum (C) obtained 15 days after the first, second and third immunization doses were serially diluted, beginning at 1:200, and used in an ELISA assay. The threshold was calculated using the mean absorbance value of the blank wells plus two standard deviations. The arrows indicate antibody titer endpoints observed after immunization.
(TIF)

**S6 Fig. rSm14-specific IgG2c antibody titer endpoints in immunized mice serum.** Pools of sera from mice immunized with rSm14 + Freund's **(A)**, rSm14 + MPLA **(B)** or rSm14 + MPLA+Alum **(C)** obtained 15 days after the first, second and third immunization doses were serially diluted, beginning at 1:50, and used in an ELISA assay. The threshold was calculated using the mean absorbance value of the blank wells plus two standard deviations. The arrows indicate antibody titer endpoints observed after immunization.
(TIF)

**S1 Table. Identification of proteins by mass spectrometry.**
(DOCX)

**S2 Table. Summary of the cellular immune response triggered by the different formulations after the third dose of immunization.**
(DOCX)

**S1 File. Raw images from SDS-PAGE and Western Blot presented in the Fig 1 and S2 Fig of the manuscript.** Raw Images of the 15% SDS-PAGE **(C)** and Western blot **(D)** using a monoclonal 6x-His-tag antibody for the analysis of rSm14 expression and purification presented in Fig 1 – Molecular weight markers (lane 1); non-IPTG induced (lane 2); 1mM IPTG-induced protein expression in bacterial culture (lane 3); bacterial lysate under denaturing conditions (lane 4); and purified rSm14 fraction (lane 5). Raw image of the SDS-PAGE gels presented in S2 Fig Heat-denatured rSm14 (A) and rSm14 (B) subjected to trypsin digestion at 25°C for 0 (T0), (lane 2); 1 (T1) (lane 3); 2 (T2) (lane 4); 5 (T5) (lane 5); 10 (T10) (lane 6); 20 (T20) (lane 7); 40 (T40) (lane 8); and 60 (T60) minutes (lane 9); Molecular weight markers (lane 1).
(PDF)

## Acknowledgments

The authors would like to thank the Program for Technological Development in Tools for Health-PDTIS- FIOCRUZ for the use of its Flow Cytometry facility and to Recombinant Production Laboratory from IRR. We thank the Lobato Paraense Mollusk facility from Instituto René Rachou (IRR) – Fiocruz-MG for providing the *S. mansoni* cercariae and the Animal`s facilities of the René Rachou Institute for the provision and maintenance of experimental animals. We thank the IRR's

project support service for the support in the project management and to the GIPB Research group for the use of its laboratory infrastructure. We thank Sueleny Silva Ferreira Teixeira for her assistance in the experiments, Patrícia Martins Parreiras for providing SWAP and Luke Anthony Baton for proofreading the manuscript.

## Author contributions

**Conceptualization:** Lis Ribeiro do Valle Antonelli, Cristina Toscano Fonseca.

**Data curation:** Poliane Silva Maciel, Gregório Guilherme Almeida.

**Formal analysis:** Poliane Silva Maciel, Lis Ribeiro do Valle Antonelli, Cristina Toscano Fonseca.

**Funding acquisition:** Lis Ribeiro do Valle Antonelli, Cristina Toscano Fonseca.

**Investigation:** Poliane Silva Maciel, Gregório Guilherme Almeida, Gardênia Braz Figueiredo de Carvalho, Rosiane A. da Silva-Pereira, Lis Ribeiro do Valle Antonelli, Cristina Toscano Fonseca.

**Project administration:** Cristina Toscano Fonseca.

**Resources:** Lis Ribeiro do Valle Antonelli, Cristina Toscano Fonseca.

**Supervision:** Cristina Toscano Fonseca.

**Validation:** Poliane Silva Maciel.

**Writing – original draft:** Poliane Silva Maciel.

**Writing – review & editing:** Gregório Guilherme Almeida, Gardênia Braz Figueiredo de Carvalho, Rosiane A. da Silva-Pereira, Lis Ribeiro do Valle Antonelli, Cristina Toscano Fonseca.

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
