## [Decision Letter · Decision Letter 0]

4 Aug 2025

Dear Dr. Fonseca,

We look forward to receiving your revised manuscript.

Kind regards,

Rui Tada, Ph.D.

Academic Editor

PLOS ONE

“This study was financed in part by the Coordination for the Improvement of Higher Education Personnel (Coordenação de Aperfeiçoamento de Pessoal de Nível Superior-CAPES) – Finance Code 001, Instituto René Rachou, Conselho Nacional de Desenvolvimento Científico e Tecnológico–Brasil (Grant Nos. 306188/2022-8 and 315540/2023-0), Rede De Pesquisa em Doenças Infecciosas Humanas e Animais do Estado de Minas (RED00313-16), Rede Mineira de  Imunobiológicos (RED00067-23), Rede Mineira de Investigação em Mucosas e Pele (RED – 00096-22) and Programa de Pós-Graduação em Ciências da Saúde – IRR.”

4. We notice that your supplementary table 1 is included in the manuscript file. Please remove them and upload them with the file type 'Supporting Information'. Please ensure that each Supporting Information file has a legend listed in the manuscript after the references list.

Reviewers' comments:

Reviewer's Responses to Questions

**Comments to the Author**

1. Is the manuscript technically sound, and do the data support the conclusions?

Reviewer #1: Yes

Reviewer #2: Partly

2. Has the statistical analysis been performed appropriately and rigorously?

Reviewer #1: Yes

Reviewer #2: No

3. Have the authors made all data underlying the findings in their manuscript fully available?

Reviewer #1: Yes

Reviewer #2: No

4. Is the manuscript presented in an intelligible fashion and written in standard English?

Reviewer #1: Yes

Reviewer #2: Yes

Reviewer #1: Please, spell MPLA and explain EM: Effector memory?

Supplementary Figures 1-5 will be invaluable in the article.

Please devise a Table to summarize, abstract, compare, and contrast the final pre-challenge immune responses achieved following vaccination with the three different Sm14 formulations in order to clarify, tabulate and emphasize all that is non protective even following challenge with 30 cercariae.

Reviewer #2: Major Comments:

1. Lack of Protective Efficacy Despite Strong Immune Response:

• Although robust immune responses (e.g., IFN-γ, TNF, IgG2c, memory T/B cells) were induced, none of the vaccine formulations resulted in significant reduction in worm burden or egg count. This raises serious concerns about the functional relevance of these immune responses.

• The discussion hypothesizes potential causes (e.g., improper folding of rSm14 due to denaturation), but these claims remain speculative. The authors should consider validating this by testing conformational integrity (e.g., circular dichroism, native PAGE, or proteolytic sensitivity assays) or referencing studies that confirm folding status under their production conditions.

2. Use of Freund’s Adjuvant in Vaccine Trials:

• While Freund’s adjuvant is commonly used in experimental settings, it is not suitable for clinical translation due to safety concerns. The authors acknowledge this but still heavily rely on data from Freund’s-formulated groups. Greater emphasis should be placed on clinically relevant adjuvants like MPLA/Alum.

3. Statistical Power and Interpretation:

• The study uses groups of 10 mice (with pooled blood samples for some flow cytometry analyses). While this is acceptable, individual-level data (e.g., scatter plots) for parasite burden and egg counts would strengthen the analysis and transparency of variance.

• The p-values for worm and egg burden are generally >0.05, yet the manuscript refers to “trends.” PLOS ONE requires data to support claims robustly—avoid overinterpreting nonsignificant results.

4. Antibody Functionality Not Adequately Addressed:

• Although anti-rSm14 antibodies recognized SWAP, it is unclear if these antibodies are functionally protective (e.g., neutralizing, opsonizing, complement-fixing). A functional assay such as in vitro parasite killing or opsonophagocytosis would be valuable, or at least a clearer discussion of this limitation.

5. Ethics and Methodology Reporting:

• Ethical approval is mentioned appropriately (license numbers provided).

• However, key methodological details—such as the justification for 30 vs. 100 cercariae challenge in different trials, and reproducibility between batches—need clarification.

⸻

Minor Comments:

• Typographical and Language Issues:

• Example: Line 16 – “described” should be “describe.”

• Line 430 – “native ,” spacing error.

• Please review the manuscript thoroughly for minor typographic inconsistencies.

• Figures & Supplementary Files:

• Ensure that all figure legends are complete and informative.

• High-resolution images and fully labeled flow cytometry panels are critical for peer review. Please double-check Supplementary Figures S1–S5 for completeness.

• Data Availability Statement:

• Authors claim all data are available within the manuscript and supplements. Please ensure raw flow cytometry data or ELISA plate readers are available upon request or in a public repository as per PLOS Data Policy.

**Do you want your identity to be public for this peer review?** For information about this choice, including consent withdrawal, please see our Privacy Policy

Reviewer #1: **Yes: ** Rashika El Ridi

Reviewer #2: **Yes: ** Mohamed Samy Abousenna

---

## [Author Response · Author response to Decision Letter 1]

10 Oct 2025

Reviewer #1: Please, spell MPLA and explain EM: Effector memory?

Response: MPLA spell was included in the abstract, page 1, line 18. EM stands for Effector memory, this information is included in the text at page 12, line 250

Supplementary Figures 1-5 will be invaluable in the article.

Response: We greatly appreciate the reviewer’s thoughtful suggestion. While we recognize the value of including these figures in the main text, we have opted to limit the number of figures. Considering that the data are already presented in the manuscript in a different format, we have chosen to keep them as supplementary figures. We believe this approach preserves the clarity and conciseness of the main manuscript while still making the data fully accessible to readers.

Please devise a Table to summarize, abstract, compare, and contrast the final pre-challenge immune responses achieved following vaccination with the three different Sm14 formulations in order to clarify, tabulate, and emphasize all that is non-protective even following challenge with 30 cercariae.

Response: We appreciate the reviewer's suggestion. Indeed, summarizing the findings related to the immune response elicited by the vaccine formulations after the third dose is a strategy that enhances the overall understanding of the results. Instead of presenting the data in a table, we chose to do so in a more visual manner through a graphical abstract, which has been included as Figure 7 in the article.

Reviewer #2: Major Comments:

1. Lack of Protective Efficacy Despite Strong Immune Response:

• Although robust immune responses (e.g., IFN-γ, TNF, IgG2c, memory T/B cells) were induced, none of the vaccine formulations resulted in significant reduction in worm burden or egg count. This raises serious concerns about the functional relevance of these immune responses.

Response: That is precisely the point we intended to highlight with our manuscript. As discussed in the manuscript (page 20, lines 445-449), the production of IgG antibodies and subclasses, as well as cytokines such as IFN-γ, following immunizations, is currently being assessed in Phase 2 clinical trials using rSm14 as the antigen. Although Phase 2 trials do not allow a direct correlation between resistance and immunogenicity, the authors suggest that the production of these factors could be associated with resistance to infection, as this association has been demonstrated in preclinical studies. However, it is important to note that in preclinical studies, a different form of rSm14 (fused with MBP) was used as the antigen.

• The discussion hypothesizes potential causes (e.g., improper folding of rSm14 due to denaturation), but these claims remain speculative. The authors should consider validating this by testing conformational integrity (e.g., circular dichroism, native PAGE, or proteolytic sensitivity assays) or referencing studies that confirm folding status under their production conditions.

Response: We thank the referee for the suggestion. Although it was not possible to perform circular dichroism, we conducted a proteolysis susceptibility assay to assess the protein folding. The results, included on page 10-11, lines 218 to 220, demonstrate that the protein is less susceptible to proteolysis than its heat-denatured form. These findings suggest the acquisition of a stable structure with fewer exposed cleavage sites. Together with the data showing recognition of the native protein in SWAP by anti-rSm14 antibodies, there is evidence that the protein’s conformational structure is correct; however, additional experiments would be needed to confirm this. These details have been added to the manuscript page 22 lines 479-487.

2. Use of Freund’s Adjuvant in Vaccine Trials:

• While Freund’s adjuvant is commonly used in experimental settings, it is not suitable for clinical translation due to safety concerns. The authors acknowledge this but still heavily rely on data from Freund’s-formulated groups. Greater emphasis should be placed on clinically relevant adjuvants like MPLA/Alum.

Response: As suggested by referee, we reduced the focus given to Freund's adjuvant in the manuscript

3. Statistical Power and Interpretation:

• The study uses groups of 10 mice (with pooled blood samples for some flow cytometry analyses). While this is acceptable, individual-level data (e.g., scatter plots) for parasite burden and egg counts would strengthen the analysis and transparency of variance.

Response: The use of pooled blood samples was necessary to obtain the number of cells required for the flow cytometry experiments, whereas parasite and egg counts were analyzed at the individual level. As suggested by the referee, we replaced Table 2 by Figure 6, which shows scatter plots of these results.

• The p-values for worm and egg burden are generally >0.05, yet the manuscript refers to “trends.” PLOS ONE requires data to support claims robustly—avoid overinterpreting nonsignificant results.

Response: We changed the description of these results. Additionally, we removed from the manuscript all text that was overinterpreting nonsignificant results.

4. Antibody Functionality Not Adequately Addressed:

• Although anti-rSm14 antibodies recognized SWAP, it is unclear if these antibodies are functionally protective (e.g., neutralizing, opsonizing, complement-fixing). A functional assay such as in vitro parasite killing or opsonophagocytosis would be valuable, or at least a clearer discussion of this limitation.

Response: As suggested, we highlighted the limitation of not having assessed the protective functionality of the antibodies raised against Sm14 in page 22 lines 483-486.

5. Ethics and Methodology Reporting:

• Ethical approval is mentioned appropriately (license numbers provided).

• However, key methodological details—such as the justification for 30 vs. 100 cercariae challenge in different trials, and reproducibility between batches—need clarification.

Response: As suggested, we included the justification of using 30 and 100 cercariae challenge in different trials page 17 lines 389-392 . Regarding protein batch, the same batch was used in all immunization trials, this information was added to the manuscript, page 6 line 109.

Minor Comments:

• Typographical and Language Issues:

• Example: Line 16 – “described” should be “describe.” - done

• Line 430 – “native ,” spacing error – done

• Please review the manuscript thoroughly for minor typographic inconsistencies -done

• Figures & Supplementary Files:

• Ensure that all figure legends are complete and informative - done

• High-resolution images and fully labeled flow cytometry panels are critical for peer review. Please double-check Supplementary Figures S1–S5 for completeness - done

• Data Availability Statement:

• Authors claim all data are available within the manuscript and supplements. Please ensure raw flow cytometry data or ELISA plate readers are available upon request or in a public repository as per PLOS Data Policy - raw flow cytometry data and Elisa plate readers are in the ArcaDados public repository under https://doi.org/10.35078/BADAEY. This information was added in the manuscript

---

## [Decision Letter · Decision Letter 1]

20 Oct 2025

Dear Dr. Fonseca,

Thank you for submitting your manuscript to PLOS ONE. After careful consideration, we feel that it has merit but does not fully meet PLOS ONE’s publication criteria as it currently stands. Therefore, we invite you to submit a revised version of the manuscript that addresses the points raised during the review process.

We look forward to receiving your revised manuscript.

Kind regards,

Rui Tada, Ph.D.

Academic Editor

PLOS ONE

Journal Requirements:

Reviewers' comments:

Reviewer's Responses to Questions

**Comments to the Author**

Reviewer #1: All comments have been addressed

Reviewer #2: All comments have been addressed

2. Is the manuscript technically sound, and do the data support the conclusions?

Reviewer #1: Yes

Reviewer #2: Yes

3. Has the statistical analysis been performed appropriately and rigorously?

Reviewer #1: Yes

Reviewer #2: No

4. Have the authors made all data underlying the findings in their manuscript fully available?

Reviewer #1: Yes

Reviewer #2: Yes

5. Is the manuscript presented in an intelligible fashion and written in standard English?

Reviewer #1: Yes

Reviewer #2: Yes

Reviewer #1: The data are very important and very well illustrated. Yet, please check the clarity of all Figures. Notably, Figure 1E and Figure 2 are blurry.

Reviewer #2: Dear Authors,

The revised manuscript represents a major improvement and now provides a clear, well-structured analysis of the immune responses induced by rSm14 vaccine formulations. You have addressed nearly all reviewer comments thoroughly, added new experimental validation for protein folding, and enhanced data transparency and statistical rigor.

Your work makes an important contribution to understanding why robust humoral and cellular responses to Sm14 do not necessarily correlate with protective efficacy against Schistosoma mansoni.

To further strengthen clarity, reproducibility, and impact, I suggest the following minor but valuable revisions:

⸻

Major comments

1. Protein structural integrity – show proteolysis assay data

• The trypsin-digestion assay is a welcome addition. Please include a representative SDS-PAGE image or densitometric curve as a new supplementary figure to visually demonstrate stability of native rSm14 versus the heat-denatured control.

• Quantitative visualization would substantiate the conclusion that the protein retains conformational structure.

• (Reference: Ribeiro-Santos et al., Protein Expr Purif 2021, PMID 33743354.)

2. Antibody functionality discussion

• Since antibody functional assays (e.g., in vitro killing, complement activation, opsonophagocytosis) were not performed, please expand the Discussion with a brief paragraph describing how such assays could confirm protective relevance of anti-Sm14 antibodies in future studies.

• This addition would contextualize your acknowledgment of the limitation and strengthen translational interpretation.

• (See Hotez et al., Nat Rev Microbiol 2022, https://doi.org/10.1038/s41579-022-00699-3.)

3. Summary of immune outcomes

• Figure 7 is an effective graphical abstract, but please add a concise table summarizing the main immune parameters (cytokines, antibody subclasses, CD4⁺/CD8⁺ T-cell memory, B-cell response, and protection outcome) for the three formulations.

• Such a table will help readers quickly compare formulations quantitatively.

• (Recommended by WOAH Terrestrial Manual 2024 Ch. 1.1.6 “Vaccine evaluation”).

4. Statistical and methodological transparency

• Indicate exact p-values in figure legends or Supplementary Data (not only thresholds).

• Specify the number of biological replicates (n) per group and whether samples were pooled before analysis.

• These details align with PLOS ONE Statistical Reporting Guidelines 2025 (https://journals.plos.org/plosone/s/statistical-reporting).

Minor comments

1. Abstract: Verify wording of line 18 — “formulated with either (i) MPLA, (ii) MPLA, or (iii) Freund’s adjuvant” appears to repeat MPLA. Likely one should be “MPLA/Alum.”

2. Introduction: Consider citing Riveau et al., Vaccine 2018 (https://doi.org/10.1016/j.vaccine.2018.03.016) regarding Bilharvax® to emphasize lessons on correlates of protection.

3. Methods:

• Clarify whether the proteolysis assay was repeated (technical/biological replicates).

• Indicate detection limits (pg/mL) for each cytokine in the CBA assay.

4. Results:

• Provide geometric mean titers ± 95 % CI for antibody data if available.

• Confirm all figure axes are labeled with units and sample size (n = ?).

5. Discussion: The section on IL-6 and neutrophil recruitment could cite Lambrecht et al., Nat Rev Immunol 2024 (https://doi.org/10.1038/s41577-024-00928-8) for updated mechanisms of alum-induced IL-1β/IL-18 signaling.

6. Language and style: A final proofreading pass is recommended (spacing before reference brackets, e.g., line 37 “public health [1]”).

**Do you want your identity to be public for this peer review?** For information about this choice, including consent withdrawal, please see our Privacy Policy

Reviewer #1: **Yes: ** Rashika El Ridi

Reviewer #2: **Yes: ** Mohamed samy abousenna

---

## [Author Response · Author response to Decision Letter 2]

18 Nov 2025

Journal Requirements:

Although referee suggested us to cite some works, we couldn't find the articles mention to verify if they were relevant to the manuscript. So any additional reference recommended by referee was added.

New references were added to reference list, cause we added a paragraph in the discussion section. these references were mentioned in the rebuttal letter

Response to Referee’s comments PONE-D-25-18857

Dear Editor and Reviewer,

Please find below our point-by-point responses to each of the reviewers’ comments.

Major comments

1. Protein structural integrity – show proteolysis assay data

• The trypsin-digestion assay is a welcome addition. Please include a representative SDS-PAGE image or densitometric curve as a new supplementary figure to visually demonstrate stability of native rSm14 versus the heat-denatured control.

• Quantitative visualization would substantiate the conclusion that the protein retains conformational structure.

• (Reference: Ribeiro-Santos et al., Protein Expr Purif 2021, PMID 33743354.)

Response: I believe there might be a misunderstanding, as both the SDS-PAGE image and the corresponding densitometric curve were already included as Supplementary Figure 2 in the previous revision. Additionally, the raw SDS-PAGE images are provided in the “S1 Raw Images” file. We also tried to check the reference mentioned by the referee but could not locate it in PubMed — it may contain a typographical error, the PMID does not correspond to an article published in Protein Expr. Purif journal.

2. Antibody functionality discussion

• Since antibody functional assays (e.g., in vitro killing, complement activation, opsonophagocytosis) were not performed, please expand the Discussion with a brief paragraph describing how such assays could confirm protective relevance of anti-Sm14 antibodies in future studies.

• This addition would contextualize your acknowledgment of the limitation and strengthen translational interpretation.

• (See Hotez et al., Nat Rev Microbiol 2022, https://doi.org/10.1038/s41579-022-00699-3.)

Response: We attempted to retrieve the article mentioned by the reviewer, but it does not correspond to the doi or author plus journal plus year mentioned, it may contain a typographical error, nevertheless a new paragraph has been included in the Discussion (lines 418–424), citing studies that describe antibody-mediated immune mechanisms involved in vaccination-induced parasite killing, as well as the articles reporting these functional assays. In the case of Sm14, the precise mechanisms by which antibodies mediate parasite killing are not yet fully elucidated, and no specific functional assays have been developed so far.

New references were cited in this part of the manuscript

32. Xu C ‐B, Verwaerde C, Grzych J ‐M, Fontaine J, Capron A. A monoclonal antibody blocking the Schistosoma mansoni 28‐kDa glutathione S‐transferase activity reduces female worm fecundity and egg viability. Eur J Immunol. 1991;21: 1801–1807.

33. Grezel D, Capron M, Grzych J ‐M, Fontaine J, Lecocq J ‐P, Capron A. Protective immunity induced in rat schistosomiasis by a single dose of the Sm28GST recombinant antigen: Effector mechanisms involving IgE and IgA antibodies. Eur J Immunol. 1993;23: 454–460.

3. Summary of immune outcomes

• Figure 7 is an effective graphical abstract, but please add a concise table summarizing the main immune parameters (cytokines, antibody subclasses, CD4⁺/CD8⁺ T-cell memory, B-cell response, and protection outcome) for the three formulations.

• Such a table will help readers quickly compare formulations quantitatively.

• (Recommended by WOAH Terrestrial Manual 2024 Ch. 1.1.6 “Vaccine evaluation”).

Response: As requested by the referee, a summary table was included as Supplementary table 2 (S2 Table)

4. Statistical and methodological transparency

• Indicate exact p-values in figure legends or Supplementary Data (not only thresholds).

• Specify the number of biological replicates (n) per group and whether samples were pooled before analysis.

• These details align with PLOS ONE Statistical Reporting Guidelines 2025 (https://journals.plos.org/plosone/s/statistical-reporting).

Response: The exact p value was included in the figures. And the number of biological replicates (n) was added to each figure’s legend.

Minor comments

1. Abstract: Verify wording of line 18 — “formulated with either (i) MPLA, (ii) MPLA, or (iii) Freund’s adjuvant” appears to repeat MPLA. Likely one should be “MPLA/Alum.” Done

2. Introduction: Consider citing Riveau et al., Vaccine 2018 (https://doi.org/10.1016/j.vaccine.2018.03.016) regarding Bilharvax® to emphasize lessons on correlates of protection.

Response: The reference Riveau et al., Vaccine 2018 is already cited in line 66 of the introduction section.

3. Methods:

• Clarify whether the proteolysis assay was repeated (technical/biological replicates).

Response: The proteolysis assay was performed twice using the same protein batch, at 25 °C and 37 °C with similar results. This information was included in the manuscript (lines 131-132).

• Indicate detection limits (pg/mL) for each cytokine in the CBA assay.

Response: The detection limits for each cytokine in CBA were added to the manuscript.

4. Results:

• Provide geometric mean titers ± 95 % CI for antibody data if available.

Response: Antibody titer was performed with pooled samples

• Confirm all figure axes are labeled with units and sample size (n = ?).

Response: Sample size was included in the figure legend.

5. Discussion: The section on IL-6 and neutrophil recruitment could cite Lambrecht et al., Nat Rev Immunol 2024 (https://doi.org/10.1038/s41577-024-00928-8) for updated mechanisms of alum-induced IL-1β/IL-18 signaling.

Response: We thank the reviewer for this suggestion. We attempted to retrieve the cited article (Lambrecht et al., Nat Rev Immunol 2024; DOI: 10.1038/s41577-024-00928-8), but this DOI does not currently correspond to a published paper in Nature Reviews Immunology or in PubMed, suggesting a possible typographical error in the reference.

6. Language and style: A final proofreading pass is recommended (spacing before reference brackets, e.g., line 37 “public health [1]”). – Response: A final proofreading was performed.

---

## [Decision Letter · Decision Letter 2]

20 Nov 2025

Exploring the immune responses triggered by vaccine formulations containing the recombinant Schistosoma mansoni 14kDa fatty acid-binding protein

PONE-D-25-18857R2

Dear Dr. Fonseca,

We’re pleased to inform you that your manuscript has been judged scientifically suitable for publication and will be formally accepted for publication once it meets all outstanding technical requirements.

Kind regards,

Rui Tada, Ph.D.

Academic Editor

PLOS ONE

Additional Editor Comments (optional):

Reviewers' comments:

Reviewer's Responses to Questions

**Comments to the Author**

Reviewer #2: All comments have been addressed

2. Is the manuscript technically sound, and do the data support the conclusions?

Reviewer #2: Yes

3. Has the statistical analysis been performed appropriately and rigorously?

Reviewer #2: Yes

4. Have the authors made all data underlying the findings in their manuscript fully available?

Reviewer #2: Yes

5. Is the manuscript presented in an intelligible fashion and written in standard English?

Reviewer #2: Yes

Reviewer #2: (No Response)

**Do you want your identity to be public for this peer review?** For information about this choice, including consent withdrawal, please see our Privacy Policy

Reviewer #2: **Yes: ** Mohamed Samy Abousenna

---

## [Editor Report · Acceptance letter]

27 Nov 2025

PONE-D-25-18857R2

PLOS ONE

Dear Dr. Fonseca,

I'm pleased to inform you that your manuscript has been deemed suitable for publication in PLOS ONE. Congratulations! Your manuscript is now being handed over to our production team.

Kind regards,

on behalf of

Dr. Rui Tada

Academic Editor

PLOS ONE